# Stable Isotopes Reveal the Dominant Species to Have the Widest Trophic Niche of Three Syntopic *Microtus* Voles

**DOI:** 10.3390/ani11061814

**Published:** 2021-06-17

**Authors:** Linas Balčiauskas, Raminta Skipitytė, Andrius Garbaras, Vitalijus Stirkė, Laima Balčiauskienė, Vidmantas Remeikis

**Affiliations:** 1Nature Research Centre, Akademijos 2, 08412 Vilnius, Lithuania; raminta.skipityte@ftmc.lt (R.S.); vitalijus.stirke@gamtc.lt (V.S.); laima.balciauskiene@gamtc.lt (L.B.); 2Center for Physical Sciences and Technology, Saulėtekio av. 3, 02300 Vilnius, Lithuania; andrius.garbaras@ftmc.lt (A.G.); vidmantas.remeikis@ftmc.lt (V.R.)

**Keywords:** small herbivores, voles, niche width, orchards, berry plantations, meadows, Lithuania

## Abstract

**Simple Summary:**

Diets and the trophic positions of animals are fundamental issues in their ecology. We analysed the isotopic niches (as a proxy for trophic niches) of common (*Microtus arvalis*), field (*M. agrestis*), and root (*M. oeconomus*) voles co-occurring in orchards, berry plantations, and nearby meadows using isotopic (*δ*^15^N and *δ*^13^C) compositions from hair samples. We tested if the niche of the dominant common vole was widest, whether its width was related to the presence of other *Microtus* species, and whether there were intraspecific differences in average *δ*^13^C and *δ^1^*^5^N stable isotope values. The obtained results showed relative stability in the trophic niche across the vegetative period. The isotopic niche of the common vole was the widest, exceeding the other two *Microtus* species by 1.6–3 times. Co-occurring vole species were separated according to *δ*^13^C (i.e., used different plants as main food), but they maintained similarity according to *δ*^15^N distribution. The effect of animal age and gender on the width of the trophic niche was strongest in root vole, which is a species that has spread across the country in the last 70 years. These results give new insights into the trophic ecology small herbivores, showing the impact of species co-occurrence.

**Abstract:**

Diets and trophic positions of co-occurring animals are fundamental issues in their ecology, and these issues in syntopic rodents have been studied insufficiently. Using carbon (*δ*^13^C) and nitrogen (*δ*^15^N) stable isotope ratios from hair samples, we analysed the trophic niches of common (*Microtus arvalis*), field (*M. agrestis*), and root (*M. oeconomus*) voles co-occurring in orchards, berry plantations, and nearby meadows (as control habitat to orchards and plantations). We tested if the niche of the dominant common vole was the widest, whether its width depended on the presence of other vole species, and whether there were intraspecific differences. Results suggest stability in the trophic niches of all three *Microtus* species, as season explained only 2% of the variance. The widest trophic niche was a characteristic of the dominant common vole, the range of *δ*^13^C values exceeding the other two species by 1.6, the range of *δ*^15^N values exceeding the other two species by 1.9, and the total area of niche exceeding that of the other voles by 2.3–3 times. In the meadows and apple orchards, co-occurring vole species were separated according to *δ*^13^C (highest values in the dominant common vole), but they maintained similar *δ*^15^N values. Results give new insights into the trophic ecology small herbivores, showing the impact of species co-occurrence.

## 1. Introduction

Animal diets and their trophic positions are fundamental issues in ecology; therefore, they are studied through a number of different methods [1,2]. The variety of diets in rodents [3,4,5] depends, among other factors, on the sympatry and syntopy of their species and the diversity of inhabited habitats [6]. While investigations into the diet of sympatric rodents, i.e., those occurring in the same region, are quite common, analysis of the diet of syntopic species (those using the same habitat, sensu Hart et al. [7]) is not (see [8]).

In the temperate climate zone, most rodent species belong to one of three groups according to their diet: granivores, herbivores (*Microtus* and *Arvicola* species), or omnivores [9].

Our study concentrated on the evaluation of the trophic niche of three *Microtus* vole species, being sympatric and syntopic at the same time [10,11]. All of these species are herbivores [12,13,14,15], though food of animal origin can constitute up to 4.5% of volume or 12.3% of frequency in the diet of the field vole [16]. The foraging ecology of small herbivores has been less intensively studied [17,18], as studies of herbivory have mainly been based on data from ungulates, mostly cervids (i.e., [19]).

However, voles are territorial non-ruminants [17], inhabiting a variety of habitats, including those under anthropogenic influence [20,21] (and references therein). As they use dietary items of both agricultural and natural origin, the trophic structure in mammals inhabiting human modified landscapes may be altered [22]. In our case, all of the investigated vole species use moderately anthropogenic habitats, such as grazed grasslands, orchards, and suburban gardens [11,23,24]. It is known that dietary specialists are more sensitive to land use than generalists, but in all cases, the responses of the species are mostly negative [25]. Not all species in the different taxonomic groups are equally able to naturalize in the changed environments; therefore, they may have different patterns of abundance, diet, breeding, etc. [26].

In terms of animal niche, three primary axes are defined: time, space, and resources [27,28,29]. In co-occurring syntopic vole species, two of these dimensions (time and space, as voles were trapped in the same trapping sessions) overlap; therefore, we should expect resource partitioning [10]. Anthropogenic impact alters spatial and temporal niches, but it is not clear how changes in the diet and resource partitioning are affected, [29]. Shifts of trophic niches in relation to human-induced changes require further investigations [30]. In the case of *Microtus* voles, niche research has so far mostly targeted the temporal and spatial niche elements [31,32,33,34,35]. Therefore, we oriented to the evaluation of the trophic niche in agricultural habitats, considering also the opinion of Bolnick et al. [36] that ecological release from interspecific competition can lead to increases in niche width.

First used in diet studies over 40 years ago (see [37,38]), stable isotope analysis has become a frequent tool in mammalian ecology, using isotope ratios of carbon (*δ*^13^C) and nitrogen (*δ*^15^N) to determine basal food resources and trophic positions [39]. This approach is powerful in its application to communities, e.g., for the analysis of resource partitioning and trophic niche dimensions [10,29,40,41,42,43]. Carbon and nitrogen stable isotope ratios allow the identification of diet changes [44] or the influence of habitat conditions [45]. Variance of stable isotope ratios offer a significant technique for estimating trophic niche widths in animals [46]. Despite some concerns expressed by [47], stable isotope ratios can be used as a proxy for trophic niche [48].

The aim of our study was to evaluate the width of the trophic niche of three sympatrically or syntopically co-occurring species of *Microtus* voles according to stable isotope (*δ*^13^C and *δ*^15^N) ratios. We tested three working hypotheses:Interspecific differences of stable isotope distribution in common (*Microtus arvalis*), field (*M. agrestis*), and root (*M. oeconomus*) voles do exist, with the trophic niche of the dominant species, the common vole, being widest,The width of the trophic niche of the dominant species is related to the presence of syntopically co-occurring *Microtus* species,Intraspecific differences of the stable isotope distribution are absent, with no differences in average carbon and nitrogen stable isotope values in males and females and between the age groups of the same species.

## 2. Material and Methods

### 2.1. Study Sites and Small Mammal Trapping

Voles were trapped at 18 study sites in Lithuania (northern Europe, 55° 19′ N, 23° 54′ E) in 2018–2020, covering agricultural habitats (apple or plum orchards, currant, raspberry, and highbush blueberry plantations) and neighboring meadows as control habitats being in the vicinity of agricultural ones. At each study site, there was one orchard or plantation and one meadow. More site and habitat details are presented in Balčiauskas et al. [11] and Stirkė et al. [24]. At four sites, common voles, field voles, and root voles lived sympatrically. At five sites, two *Microtus* species were trapped, one of these being common vole. Eight other sites were inhabited by common voles only, while no *Microtus* were trapped at one site (Figure 1). At all sites with co-occurrence, common vole strongly dominated over the other two species by numbers.

Voles were snap-trapped using the standard method, with lines of 25 traps set at 5 m intervals, exposed for three days, and checked once per day in the morning [49]. Total trapping effort was 25,503 trap days over 168 trapping sessions. We define the trapping session as a three-day trapping period in the same habitat, season, and year. Thus, four trapping sessions per site were conducted each year (one in the crops, a second in the control habitat, both carried out two times per year—summer and autumn).

Voles were identified to the species level by their teeth [50] during dissection or after cleaning the skulls with *Dermestes* beetles in the laboratory. The gender of the voles and age groups were identified at dissection. We used three age categories: adults, subadults, and juveniles. Identification of the age groups was based on the status of the sex organs, body mass, and the level of atrophy of the thymus [51]. Thymus reduction occurs with animal age, the maximum size being in juveniles, down to nearly full involution in adults [52,53,54].

### 2.2. Stable Isotope Analysis

Samples of hair for carbon and nitrogen stable isotope analysis were collected from 376 of the 509 trapped vole individuals (Table 1). All suitable field vole and root vole samples were analysed. The most numerous vole species, the common vole, was sub-sampled in 2020.

The small tuft of hair (ca 5 mm wide) was clipped from between the shoulders of each specimen and stored dry. Before analysis, the samples were weighed and packed in tin capsules. The samples of the hair were not pre-treated, as earlier, we ascertained that this procedure did not change the obtained results [55]. Dirty (covered by soil or blood) samples were washed in deionised water and methanol and then dried. Very dirty samples were discarded. Carbon and nitrogen stable isotope ratios were measured at the Center for Physical Sciences and Technology, Vilnius, Lithuania, using an elemental analyser (EA) (Flash EA1112) coupled to an isotope ratio mass spectrometer (IRMS) (Thermo Delta V Advantage) via a ConFlo III interface (EA-IRMS). Five percent of the samples were run in duplicate, and the obtained results for these samples were averaged.

As reference materials, we used Caffeine IAEA-600 (*δ*^13^C = −27.771 ± 0.043‰, *δ*^15^N = 1 ± 0.2‰), Ammonium Sulfate IAEA-N-1 (*δ*^15^N = 0.4 ± 0.2‰), and Graphite USGS24 (*δ*^13^C = −16.049 ± 0.035‰) provided by the International Atomic Energy Agency (IAEA). These standards were run every 12 samples. Repeated analysis of these reference materials gave a standard deviation of less than 0.08‰ for carbon and 0.2‰ for nitrogen (see [56]).

Carbon and nitrogen stable isotope data are reported as *δ*X values (where X represents the heavier isotope ^13^C or ^15^N) or differences from given standards, expressed in parts per thousand (‰) and are calculated according to the formula:*δ*X = [R_sample_/R_standard_ − 1] × 1000,
where R_sample_ = ^13^C/^12^C or ^15^N/^14^N of the sample, R_standard_ = ^13^C/^12^C or ^15^N/^14^N of the standard.

### 2.3. Statistical Analyses

We tested if the *δ*^15^N and *δ*^13^C values were distributed normally, using Kolmogorov–Smirnov’s D test. The distributions of both isotopes in all three vole species were normal: in common vole, *δ*^13^C (D = 0.06, *p* = 0.25) and *δ*^15^N (D = 0.06, *p* = 0.21), in field vole (D = 0.12, *p* = 0.68 and D = 0.13, *p* = 0.63, respectively), and in root vole (D = 0.11, *p* = 0.73 and D = 0.15, *p* = 0.33, respectively). Based on conformity to normal distribution, parametric tests were further applied.

The proportions and 95% confidence interval of trapping sessions (or sites) where the co-occurrence of two or all three vole species was recorded was evaluated using the Wilson method and WinPepi ver. 11.39 software (Abramson, J., Jerusalem, Izrael).

The *δ*^13^C and *δ*^15^N values in the samples were expressed as arithmetic mean ± 1 SE. ANOVA was used to find the relationship of year, season, species, age, and sex of individuals to paired *δ*^15^N and *δ*^13^C distribution, using Hotteling’s two sample T^2^ test for significance. The effect size was assessed according to values of the partial eta-squared (eta^2^), which shows the proportion of the variability in the dependent variables that is explained by the effect. The interspecific influences of species, as well as intraspecific differences (between males and females, and between the three age groups), on the carbon and nitrogen stable isotope values were tested with parametric ANOVA, using Wilk’s lambda test for significance. Differences between groups were evaluated with post-hoc Tukey test.

The positions of species and intraspecific groups, including those with sample size n < 5, in the isotopic biplot was shown using SigmaPlot ver. 12.5 (Systat Software Inc., San Jose, CA, USA). Isotopic niches of species, using parameters of TA (total area), SEA (standard ellipse area), and SEAc (as corrected central ellipses, unbiased for the sample size), were calculated using the package SIBER [57] under R ver. 3.5.0 (https://cran.r-project.org/bin/windows/base/rdevel.html, accessed on 2 March 2019). All other calculations were performed using Statistica for Windows ver. 6 (StatSoft, Inc., Tulsa, OK, USA).

## 3. Results

### 3.1. Distribution of Co-Occurring Microtus Populations

We tested if the investigated three *Microtus* vole species were syntopic, analysing their co-occurrence in all trapping sessions. In 46.4% (95% CI = 39.1–54.0%) of sessions, *Microtus* species were not trapped (Table 2). Common vole, as a single *Microtus* species, was trapped in 36.9% (CI = 30.0–44.4%) of sessions, field vole in 3.0% (1.3–6.8%), and root vole in 4.8% (2.4–9.1%). Two of the three species were co-trapped in 8.9% (5.5–14.2%) of sessions. All of these co-trappings were in the apple orchards or control habitats (Table 2). In the plum orchards and currant plantations, only common voles were present, while in the raspberry plantations, single individual of the root vole was trapped in addition to common voles (Table 2).

Sympatric co-occurrence (the same site, but not necessary the same habitat) was more frequent: at 22.2% (CI = 9.0–45.2%) of sites, common voles, field voles, and root voles occurred in sympatry, and at 27.8% (12.5–50.9%) of sites, two species co-occurred.

### 3.2. Interspecific Differences in Dietary Space between Three Microtus Vole Species

The species-related distribution of *δ*^13^C and *δ*^15^N values is presented in Table 3. Irrespective of other factors, the widest trophic niche was found in the common vole, exceeding other two species in the ranges of both carbon (δ^13^C) and nitrogen (δ^15^N) stable isotope ratios. The ranges of the other two species, field vole and root vole, were similar. Therefore, our first hypothesis was confirmed.

We tested the influence of the year and season on the distribution of δ^13^C and δ^15^N values in the three vole species. The cumulative influence was significant on both of the stable isotope ratios, but it explained only a very small part of the variance (year: T^2^ = 0.21, F_4,740_ = 19.4, eta^2^ = 0.095; season: T^2^ = 0.02, F_2,371_ = 3.5, eta^2^ = 0.019). Given the fact that the cumulative effect of the time factor on the distribution of *δ*^13^C and *δ*^15^N values is about 10%, this showing the relative stability of the diets, we further analysed the data irrespective of year and season.

Irrespective of the habitat (Figure 2a), we found significant differences in both *δ*^13^C and *δ*^15^N average values in the hair of co-occurring *Microtus* voles (Wilks lambda = 0.89, F_4,744_ = 11.2, *p* < 0.001). Differences in *δ*^13^C (F_2,373_ = 17.33, *p* < 0.001) were better expressed than differences in *δ*^15^N (F_2,373_ = 5.36, *p* < 0.01). The highest average value of *δ*^13^C was found in the common vole, significantly exceeding that in the field vole (Tukey HSD, *p* < 0.025) and root vole (HSD, *p* < 0.001). The highest average value of *δ*^15^N in the hair of root voles significantly exceeded that in common voles (HSD, *p* < 0.05), not differing from field voles (Figure 2a).

Significant differences in both *δ*^13^C and *δ*^15^N average values in the hair of co-occurring *Microtus* voles (Wilks lambda = 0.83, F_4,232_ = 5.82, *p* < 0.001) were found in the control habitats. Differences in *δ*^13^C (F_2,117_ = 10.99, *p* < 0.001) were significant, while no interspecies differences were observed in *δ*^15^N (F_2,117_ = 1.07, *p* = 0.35). The highest average value of *δ*^13^C was found in the common vole (Figure 2b), significantly exceeding that in the field vole (Tukey HSD, *p* < 0.005).

Interspecific differences in stable isotope values were less expressed in *Microtus* voles co-occurring in the apple orchards (Wilks lambda = 0.86, F_4,252_ = 15.11, *p* < 0.001), being significant in *δ*^13^C (F_2,127_ = 8.78, *p* < 0.001) but not in *δ*^15^N (F_2,373_ = 1.71, *p* = 0.19). The highest average value of *δ*^13^C was found in the common vole, significantly exceeding that in the field vole (Tukey HSD, *p* < 0.025) but not in the root vole. Differences in *δ*^15^N were all not significant (Figure 2c).

The dietary niches of common voles, field voles, and root voles, shown as core ellipses in the isotopic space, had a certain degree of overlap (Figure 3). The widest niche was characteristic to the common vole (TA = 12.45), being three times wider than that of the field vole (TA = 4.08) and 2.3 times wider than that of the root vole (TA = 5.41). However, core areas were of the same width (SEA = 1.87, 1.71 and 2.23; SEAc = 1.90, 1.86 and 2.38, respectively). Irrespective of the habitat, the core dietary niche of the common vole was separated from that of the other two species (Figure 3a). The overlap with the core dietary niche of the field vole in the control habitats (Figure 3b) and with the root vole in the apple orchards (Figure 3c) was less than 0.5%. The core dietary niches of the root and field voles widely overlapped in all cases: 4.3% irrespective of habitat (Figure 3a), 7.2% in the control meadows (Figure 3b), and 9.3% in the apple orchards (Figure 3c).

In the plum orchards (Figure 4a) and currant plantation (Figure 4b), we trapped only common voles, while in raspberry plantations (Figure 4c), only a single root vole individual was trapped along with a number of common voles. The influence of co-occurrence with other *Microtus* species on both of the analysed stable isotope ratios in the common vole hair was significant (T^2^ = 16.18, F_2,373_ = 8.07, *p* < 0.001). However, this difference (Figure 4 compared to Figure 2) related to the distribution of δ^15^N values only, being 14.1% higher in co-occurring common voles (5.26‰, CI = 5.08–5.45‰ versus 4.61‰, 95% CI = 4.35–4.87% in common voles without co-occurring other *Microtus* species, F_1,374_ = 15.94, *p* < 0.001; Tukey HSD, *p* < 0.001). The distribution of δ^13^C values was not affected by co-occurrence (F_1,374_ = 0.30, *p* = 0.56). Average *δ*^13^C values in the hair of common voles co-occurring with other *Microtus* species (−27.14‰, CI = −27.25–−27.02‰) were very close to those of common voles without co-occurrence (−27.18‰, CI = −27.27–−27.10‰). Thus, the second hypothesis was confirmed for nitrogen but not confirmed for carbon stable isotope distribution.

### 3.3. Intraspecific Differences in Dietary Space of Microtus Voles

The intraspecific differences in the dietary space of common, field, and root voles were analysed according to age (Figure 5) and gender of the individuals (Figure 6). Irrespective of the species, the influence of age was stronger (T^2^ = 0.04, F_4,740_ = 3.7, *p* < 0.01) than that of the gender (T^2^ = 0.10, F_2,371_ = 1.90, *p* = 0.15).

In the common vole, the effect of age on the variance of both stable isotope ratios in the hair was significant (T^2^ = 0.10, F_4,600_ = 7.3, *p* < 0.001, eta^2^ = 0.095), but it explained less than 10% of variance. The effect of gender was weak (T^2^ = 0.02, F_2,301_ = 2.6, *p* = 0.07, eta^2^ = 0.017). Univariate results show the distribution of *δ*^13^C values being dependent on the animal age (F_2,302_ = 14.1, *p* < 0.001) and gender (F_1,302_ = 4.9, *p* < 0.05). The distribution of *δ*^15^N values was not dependent on the common vole age (Figure 5a) or gender (Figure 6a). The average *δ*^13^C value in juveniles of common vole was significantly less than in adults (Tukey HSD, *p* < 0.001); the difference from subadults was not significant (HSD, *p* = 0.17).

In the field vole, the effect of age and the effect of gender on the variance of both stable isotope ratios in the hair was not significant (T^2^ = 0.07, F_4,50_ = 0.42, *p* = 0.79, and T^2^ = 0.06, F_2,26_ = 0.77, *p* = 0.47, respectively). No differences between age groups (Figure 5b) or between males and females (Figure 6b) were observed.

The effect of age (Figure 5c) and gender (Figure 6c) on the variance of both stable isotope ratios in the hair was strongest in the root vole. Animal age explained 10.7% of variance in both isotope ratios (T^2^ = 0.24, F_4,66_ = 1.99, *p* = 0.11), while animal gender explained 24.9% of variance (T^2^ = 0.33, F_2,33_ = 5.63, *p* < 0.01). Univariate results show the distribution of *δ*^15^N values being dependent on the animal age (F_2,35_ = 3.88, *p* < 0.05) and gender (F_2,35_ = 8.86, *p* < 0.01), while the distribution of *δ*^13^C values was not. The average *δ*^15^N value in males of the root vole was significantly higher than in females (Tukey HSD, *p* < 0.05), and in subadult animals, it was higher than in adults (HSD, *p* = 0.10). Other pairwise differences were not significant. Therefore, the third hypothesis was partially confirmed.

## 4. Discussion

Although the isotopic niche can be a result of many ecological and environmental factors, it is synonymous with the trophic niche when primarily driven by consumer-resource interactions [58]. Our results suggest relative stability in the trophic niches of the common vole, field vole, and root vole over time, the effect of the season being not significant and explaining just 2% of the variance. The widest trophic niche was characteristic of the dominant species, the common vole, the range of *δ*^13^C values exceeding the other two species by 1.6 times, the range of *δ*^15^N values exceeding the other two species by 1.9 times, and the total area of the isotopic niche exceeding that of the other voles by 2.3–3 times. In the control habitat (meadows) and apple orchards, the co-occurring vole species were separated according to *δ*^15^N (the highest values in common vole), but they maintained similar *δ*^13^C values.

### 4.1. Trophic Niche of Microtus Voles in Different Habitats

Compared to these moderately anthropogenic habitats, the trophic niches of root and field voles in spring-flooded meadows in west Lithuania [10] were much narrower, with lower average *δ*^15^N values and higher average *δ*^13^C values. The range of *δ*^13^C values for the root vole was just 0.91‰ (average −26.0 ± 0.07‰) and the range of *δ*^15^N was 1.27‰ (5.23 ± 0.10‰), while the respective values for field vole were 0.91‰ (average −26.6 ± 0.15‰) and 1.01‰ (4.68 ± 0.29‰). In flooded forest, the root vole had even lower values of both isotope ratios, *δ*^15^N = 3.98‰ and *δ*^13^C = −29.86‰ [10]. Therefore, we confirm that the small herbivore *Microtus* voles have seen a shift of the trophic niche under the influence of anthropogenic impact upon their habitat.

Here, we fully agree with [59] that relatively low values of *δ*^15^N cannot always be related to herbivory. Relatively higher *δ*^15^N values may be related to fertilisation [55], which may be of great importance in resource-limited orchard habitats. However, greater *δ*^15^N values in field vole hair may also be related to an increase in the proportion of animal food in their diet [43,60]. Among the investigated *Microtus* voles, only the field vole was characterised as using foods of animal origin [16,61], especially in the spring and summer periods [62]. It should be noted that many other herbivore species, including common voles, are capable of omnivory to some extent [63]. In addition, field voles can exhibit preferences for food that is not common in the habitat [62].

Differences in the trophic niche of *Microtus* voles are important when choosing focal small herbivore species for pesticide risk assessment. As Lithuania is currently included within the Northern Zone, the field vole has been referred to as the representative herbivore [64]. However, as well as common vole being much more abundant and well-represented in the orchards [24], diet differences are also obvious, as shown above.

### 4.2. Possible Factors Influencing Changes in Trophic Niche

Our results are of interest for small mammal trophic ecology in general [5,18,59,65,66]. Due to agricultural activities, orchards and berry plantations are most possibly not places of abundant and diverse resources for herbivores. Grass is mowed (and removed from the orchards), while herbicides and other plant protection measures are applied [11,67,68,69], despite the knowledge that cover crops in the orchards may enable ecosystem services [70]. The limited availability of resources should lead to narrow trophic niches and reduced niche overlap in small mammals [43,66,71].

Insufficient resources of the accustomed foods also change the diets and the position of the trophic niche in other mammal groups, including carnivores [29,72]. One of the main factors responsible for niche reduction in wild mammals is human activity, creating disturbance and reducing spatial and temporal niches [73,74]. Therefore, the partitioning of the trophic niche becomes very important, allowing species to co-exist when they live in sympatry [11,29].

The role of alternative foods is also of high importance to rodents characterised by diverse diets. Diet diversity is one of the factors affecting the trophic niche for voles, especially when alternative food sources are considered [18,75].

According to [36], individual variation of the trophic niche depends on intra-individual variation (i.e., the change of individual niche breadth) and inter-individual variation (i.e., the reduction or contraction of niche overlap among individuals).

### 4.3. Role of Community and Intraspecific Patterns in Defining Trophic Niche

Our results highlight the importance of considering interspecific competition when interpreting patterns of habitat selection among coexisting species [35,36]. The common vole and field vole are morphologically similar species. They have similar ways of life [32], though they differ in their preferred habitats [76]. According to the findings of [36], a termination of interspecific competition may increase the niche width of a species, but the authors did not consider trophic niche. Based on this, syntopic populations of the three *Microtus* species should be characterised by narrowing niches in all vole species due to interspecific competition. However, this was not true: *δ*^13^C values were not affected by co-occurrence, while the range of *δ*^15^N values was significantly wider in co-occurring common voles. This finding requires further investigation, as it foresees possible species divergence in using foods of animal origin. Beyond doubt, stable isotope values indicate the trophic niches of species [48], as our samples were from the same habitats and thus not biased by habitat influence.

Bergeron and Joudoin pointed out that diet changes under interspecific competition may be very important, as diet quality is related to survivorship and health status [77]. These authors wrote that herbivores are also limited by food sources and do have different food preferences. The situation is complicated by diet dependence on population densities [77]. Wider isotopic niches are expected in human-modified landscapes [22], and this was also true for the *Microtus* voles in our study. Therefore, we see that the alteration of the trophic niche in *Microtus* voles in the orchards (agricultural habitat) could be further affected by interspecific competition between syntopic species.

### 4.4. Specificity of Agricultural Habitats to Microtus Voles

Understanding that agriculture is one of the main factors that negatively influences biological diversity, environmentally friendly farming systems in the EU were introduced from the 1980s [78]. Hedgerows, grassland inclusions, flower strips, and woodland inclusions were recognised as positive agroecological infrastructures [79]. However, investigations into the effect of environmentally friendly schemes and structures have seen a bias towards insects, birds, and other groups, but not mammals.

According to [80], 76% of publications related to biological diversity in fruit orchards were from West Europe and North America, and only 8.7% were devoted to mammals. Voles and mice in agricultural habitats are mostly treated as pests [81] (and references therein) and agrophilic rodents are mostly treated as invasive species [82]. Therefore, our approach to study the trophic niche of *Microtus* voles is rather original.

The mechanisms of coexistence of such closely related species are factors that have still not been addressed by community ecologists. Two important factors in this coexistence are, firstly, resource use and competition for these resources [83], and secondly, dietary comparisons of similar, coexisting species that can help define species niches [84]. We understand that competition for resources may be reduced due to the density-dependent selection of habitat, thus increasing the chances of co-existence [85]. However, even with three years of study at 18 sites, our materials are insufficient for such analysis. While food preferences may affect the distribution of granivores and herbivores in the agrolandscape [86], no such studies concerning syntopic herbivores have been conducted so far.

Generalising, trophic interactions may shape rodent population dynamics in resource-abundant landscapes [87], including even their number outbreaks [32,88]. In resource-poor habitats, choice of plant species is foreseen in *Microtus* [89]. However, a lack of multi-species diet investigations in agricultural habitats, let alone orchards, prevents the comparing of our conclusions regarding habitat influence on syntopic *Microtus* trophic niche with other results.

## 5. Conclusions

Shown on sympatric and syntopic *Microtus* voles, our results highlight the importance of interspecific competition for interpreting patterns of habitat selection and resource sharing among coexisting herbivore species.The widest trophic niche was characteristic to the dominant species, common vole.In the case of co-occurrence with other *Microtus* species, the width of the trophic niche of the common vole increased, separating the species according to *δ*^15^N values.Intraspecific differences in the dietary space were best expressed in the dominant common vole (differences according to *δ*^13^C, but not *δ*^15^N values) and in the root vole (according *δ*^15^N values).

## Figures and Tables

**Figure 1 animals-11-01814-f001:**
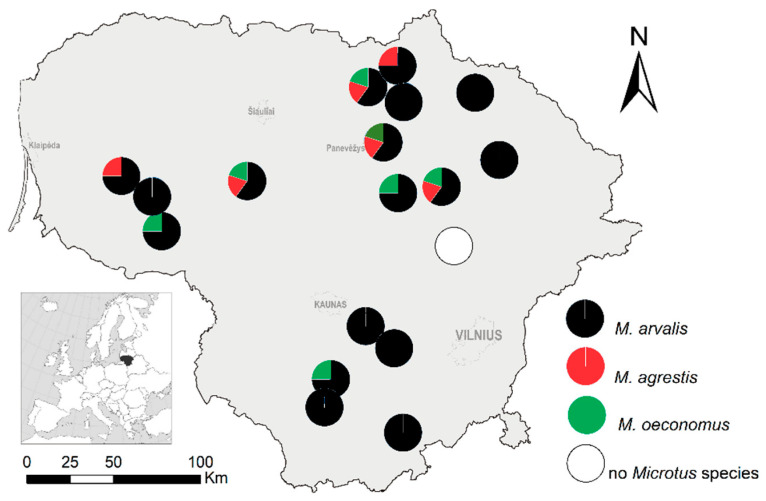
Distribution of sympatric *Microtus* species at the trapping sites in Lithuania, 2018–2020. Due to small numbers of trapped field and root voles, the proportions of these species do not correspond to the width of slices in the pie charts.

**Figure 2 animals-11-01814-f002:**
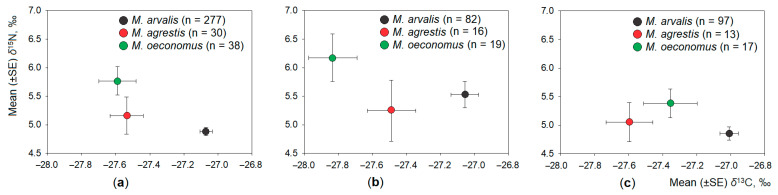
Distribution of syntopic *Microtus* species according to stable isotope ratios: (**a**) irrespective of habitat; (**b**) in control habitats (meadows); and (**c**) in apple orchards. Sample size is shown in the legend.

**Figure 3 animals-11-01814-f003:**
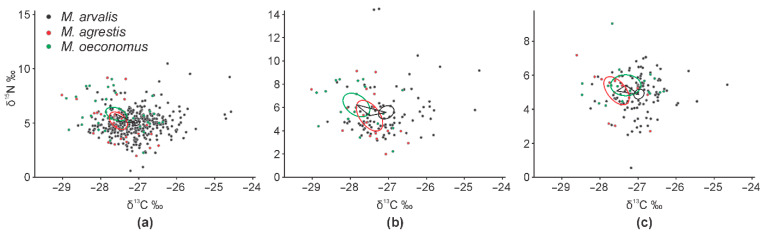
Central ellipses of syntopic *Microtus* species in the isotopic space, representing fundamental niches: (**a**) irrespective of habitat; (**b**) in control habitats; and (**c**) in apple orchards. Sample size is the same, as in Figure 2.

**Figure 4 animals-11-01814-f004:**
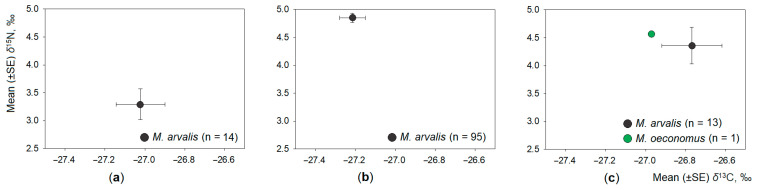
Central position (mean ± SE) of stable isotope ratios of the common vole without or in limited co-occurrence: (**a**) plum orchards; (**b**) currant plantations; (**c**) raspberry plantations. Sample size is shown in the legend.

**Figure 5 animals-11-01814-f005:**
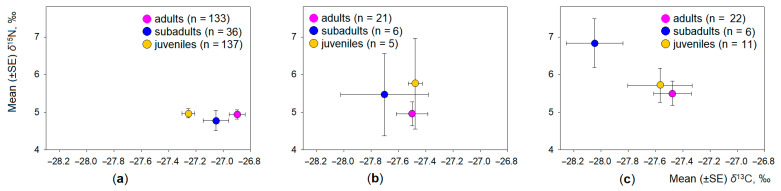
Intraspecific differences in the stable isotope values in the hair of adult, subadult, and young animals: (**a**) common vole; (**b**) field vole; (**c**) root vole. Sample size is shown in the legend.

**Figure 6 animals-11-01814-f006:**
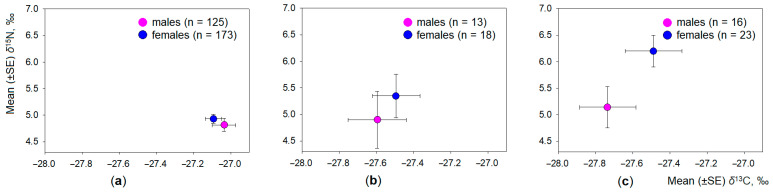
Intraspecific differences in the stable isotope values in the hair of males and females: (**a**) common vole; (**b**) field vole; (**c**) root vole. Sample size is shown in the legend.

**Table 1 animals-11-01814-t001:** Samples of *Microtus* voles from commercial orchards of Lithuania, 2018–2020, used for stable isotope analysis.

Species	Trapped	Analysed	Adults	Sub-Adults	Juveniles	Males	Females
*M. arvalis*	436	306	133	36	137	129	177
*M. agrestis*	31	31	21	6	4	13	18
*M. oeconomus*	42	39 *	22	6	11	16	23

* Some individuals were partially destroyed; therefore, their hair was not suitable.

**Table 2 animals-11-01814-t002:** Co-occurring of three *Microtus* species. N—number of trapping sessions (different years, seasons, and habitats), NONE—no *Microtus* trapped, CV—common vole only, FV—field vole only, RV—root vole only, and combinations of these species, n—number of sites with the habitat present.

Habitat	N	NONE	CV	FV	RV	CV + FV	CV + RV	FV + RV
Apple orchards (n = 10)	46	21	15	1	2	2	4	1
Plum orchards (n = 2)	8	3	5					
Raspberry plantations (n = 3)	15	7	7		1			
Currant plantations (n = 3)	14	5	9					
High blueberry plantations (n = 1)	3	3						
Control meadows (n = 16)	82	39	26	4	5	1	6	1

**Table 3 animals-11-01814-t003:** Central position (mean ± SE) and ranges of stable isotope ratios in the hair of three *Microtus* species from commercial orchards of Lithuania, 2018–2020.

Species	δ^13^C Values, ‰			δ^15^N Values, ‰		
	Mean ± SE	Min–Max	Range	Mean ± SE	Min–Max	Range
*M. arvalis*	−27.08 ± 0.04	−28.58–−24.58	4.00	4.93 ± 0.08	0.52–14.50	13.98
*M. agrestis*	−27.54 ± 0.10	−29.00–−26.48	2.52	5.16 ± 0.33	1.91–9.13	7.22
*M. oeconomus*	−27.59 ± 0.11	−28.88–−26.37	2.51	5.77 ± 0.25	2.16–9.01	6.85

## Data Availability

After publication, research data will be available from the corresponding author upon request. The data are not publicly available due to usage in the ongoing study.

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
