# Peer review of "Stable Isotopes Reveal the Dominant Species to Have the Widest Trophic Niche of Three Syntopic Microtus Voles"

_animals, 2021, doi:10.3390/ani11061814_

Round 1
Reviewer 1 Report
This paper is very well written and a pleasure to read.
Line 97: This is not a complete sentence. Maybe it would be better to say something like " Site details can be found in previous published study by Stirke et al (24).
Line 104: Make sure to include the white circle in the key that indicates no voles. Or just omit it from map and key.
Line 127: Please add how much hair was sampled.
Line 128: I am not clear what the term scissored is referring to. Do you mean that the hair was further cut into smaller pieces?
Line 184: Table 2 In habitat i would add how many apple orchards were trapped in eg Apple Orchard (n?). I would also label control as meadows if that is what they all are. I am not sure why they are considered controls? This might not be the best term to describe them
Author Response
Rev#1 comments
This paper is very well written and a pleasure to read.
Answer: thank you very much for the positive evaluation of our manuscript. Answer to your comments are presented below.
Comment: Line 97: This is not a complete sentence. Maybe it would be better to say something like " Site details can be found in previous published study by Stirke et al (24).
Answer: we expanded this part of text, as shown below.
Voles were trapped at 18 study sites in Lithuania (northern Europe, 55° 19’ N, 23° 54’ E) in 2018–2020, covering agricultural habitats (apple or plum orchards, currant, raspberry, and highbush blueberry plantations) and neighboring meadows as control habitats being in the vicinity of agricultural ones. At each study site there was one orchard or plantation and one meadow. More site and habitat details are presented in Balčiauskas et al., [11] and Stirkė et al., [24].
Comment: Line 104: Make sure to include the white circle in the key that indicates no voles. Or just omit it from map and key.
Answer: We did as advised.
Comment: Line 127: Please add how much hair was sampled.
Answer: in the process of sampling, we clipped small tuft (no more than 5 mm width at the hair base), though later not all this hair were used. Text changed:
The small tuft of hair (ca 5 mm wide) was clipped from between the shoulders of each specimen and stored dry.
Comment: Line 128: I am not clear what the term scissored is referring to. Do you mean that the hair was further cut into smaller pieces?
Answer: thank you for pointing this out. In fact, only first sample were scissored, i.e., cut into two parts, as we used small tin capsules and long hair did not fit. Later, however, we used 8 mm capsules, therefore there was no need to scissor hair. Both types of capsules are standard and the size has no influence to results.
Therefore, answering your comment we deleted the word “scissored”.
Comment: Line 184: Table 2 In habitat i would add how many apple orchards were trapped in eg Apple Orchard (n?). I would also label control as meadows if that is what they all are. I am not sure why they are considered controls? This might not be the best term to describe them
Answer: control areas were designated at each orchard or berry plantation; they were not subjected to agricultural activities applied to orchards and plantations, therefore named control habitats. Accepting your comment, we changed “control habitats” to “control meadows” and added number of all habitats to the first column of the Table 2. Text “n – number of sites with the habitat present” was added to the Table caption.
Reviewer 2 Report
This is a very nice paper and I read it with interest. I like this approach to testing inter and intraspecific competition. I am convinced that this paper is worth publishing but what should help to understand it better is more detailed and clearer methods description. And this what most of my remarks refer to, please see below.
Simple summary – I think a sentence or two on the principles of using stable isotopes to study diet of animals would give a nice background and help to understand the whole experiment.
Abstract – “In the control habitat (meadows) and apple orchards…” It is not clear, a control habitat to what? To me, both simple summary and abstract lack a sentence that would show the main conclusion.
Key words – why are only orchards mentioned here? what about berries plantations and meadows?
line 65: two of these dimensions overlap, there- 65 fore we should expect resource partitioning – please specify which dimensions
line 80-81 – this sentence is quite awkward and difficult to understand at the first glance, please try to rephrase.
lines 85-92 – I do feel that (similarly as in the abstract) not enough background was given. When you refer to dominant species which one is meant? To me these hypotheses are not well supported by the introduction.
line 95-96 – this is not clear to me, in each study site there were berries plantations, orchards and meadows, so there were like three localities in each study site?
line 97 - Site details are presented in [24] – I would like to see some site description here, at least brief one, but this would help to understand the study design.
line 104 - proportions of 104 individuals of these species are not reflected exactly) – what this means?
line 109 - Thus, at least four trapping sessions per site – there could be more than four? And does this mean than all plantations were pooled together?
line 120 - Samples of hair for carbon and nitrogen stable isotope analysis were collected from 120 376 of the 509 trapped vole individuals – if hair was taken for analysis why such an invasive method (like snap traps and not live traps) was used? I think this should be somehow justified as in many countries this would be difficult to force.
line 127 – maybe it is worth saying how much hair was taken for analysis, this may be important for someone planning similar research but with live trapping
line 179 - All of these co-trappings were in the apple orchards or control habitats – how about berries plantations? I guess with no clear description of the study sites and the whole study design it is difficult to understand results.
line 181 - At the site level,.. – what do you understand by ‘the site level’?
Table 2 – so all of those habitats were present at each trapping site? with the same number of traps for each habitats and the same trapping effort?
Figure 2 – ok, here you refer to apple orchards, do this means that plantations were not treated together?; it should be said what were control habitats (e.g. in meadows put in brackets)
Author Response
Rev#2 comments
This is a very nice paper and I read it with interest. I like this approach to testing inter and intraspecific competition. I am convinced that this paper is worth publishing but what should help to understand it better is more detailed and clearer methods description. And this what most of my remarks refer to, please see below.
Answer: thank you very much for the positive evaluation of our manuscript. Answer to your comments are presented below.
Comment: Simple summary – I think a sentence or two on the principles of using stable isotopes to study diet of animals would give a nice background and help to understand the whole experiment.
Answer: we are sorry, in the simple summary there is no more space for the background; problem is addressed in the very first sentence. According the journal requirements, “The simple summary consists of no more than 200 words in one paragraph and contains a clear statement of the problem addressed, the aims and objectives, pertinent results, conclusions from the study” – we got an impression that background is not foreseen. We changed the concluding (last) sentence of the simple summary: “These results give new insights into the trophic ecology small herbivores, showing impact of species co-occurrence.”
Comment: Abstract – “In the control habitat (meadows) and apple orchards…” It is not clear, a control habitat to what? To me, both simple summary and abstract lack a sentence that would show the main conclusion.
Answer: meadows were used as control habitat for agricultural ones (orchards and plantations). Text changed, now is
Using carbon (δ13C) and nitrogen (δ15N) stable isotope ratios from hair samples, we analysed the trophic niches of common (Microtus arvalis), field (M. agrestis) and root (M. oeconomus) voles co-occurring in orchards, berry plantations and nearby meadows (as control habitat to orchards and plantations).
Comment: Key words – why are only orchards mentioned here? what about berries plantations and meadows?
Answer: we added suggested keywords
Keywords: small herbivores; voles; niche width; orchards; berry plantations; meadows; Lithuania
Comment: line 65: two of these dimensions overlap, therefore we should expect resource partitioning – please specify which dimensions
Answer: sentence changed; syntopic co-occurring means, that different vole species were present in the same time and in the same habitat. Unfortunately, we may judge about resources only from the analysis of stable isotopes. Text now is:
In co-occurring syntopic vole species, two of these dimensions (time and space, as voles were trapped in the same trapping sessions) overlap, therefore we should expect resource partitioning [10].
Comment: line 80-81 – this sentence is quite awkward and difficult to understand at the first glance, please try to rephrase.
Answer: changed to “Despite some concerns expressed by [47], stable isotope ratios can be used as a proxy for trophic niche [48].”
Comment: lines 85-92 – I do feel that (similarly as in the abstract) not enough background was given. When you refer to dominant species which one is meant? To me these hypotheses are not well supported by the introduction.
Answer: we added text to the first sentence of Abstract to widen background presentation “Diets and trophic positions of co-occurring animals are fundamental issues in their ecology, these issues in syntopic rodents studied insufficiently.”
We also put “common vole” to indicate who is the dominant one.
Change in the text in Lines 85-92 were done to address your comment.
Comment: line 95-96 – this is not clear to me, in each study site there were berries plantations, orchards and meadows, so there were like three localities in each study site?
Answer: no, there were two habitats in one study site (we do not use word localities, as these habitats were not far apart): one orchard (or berry plantation) and one control habitat. After expanding the text in Line 97 this should be clear now – see also answer below.
Comment: line 97 - Site details are presented in [24] – I would like to see some site description here, at least brief one, but this would help to understand the study design.
Answer: Study design was: one agricultural habitat (orchard or berry plantation) and one control habitat, not subjected to the agricultural activities used in orchards or plantations. We expanded text as suggested.
Voles were trapped at 18 study sites in Lithuania (northern Europe, 55° 19’ N, 23° 54’ E) in 2018–2020, covering agricultural habitats (apple or plum orchards, currant, raspberry, and highbush blueberry plantations) and neighboring meadows as control habitats being in the vicinity of agricultural ones. At each study site there was one orchard or plantation and one meadow. More site and habitat details are presented in Balčiauskas et al., [11] and Stirkė et al., [24].
Comment: line 104 - proportions of individuals of these species are not reflected exactly) – what this means?
Answer: proportions of sympatric Microtus species were different, and there are not possible to show proportions of M. agrestis or M. oeconomus on the map, as they are very small and will not be visible. That is the reason why slices in the pie charts do not correspond to the species shares. We changed caption of the Figure 1 to make this clear.
Figure 1. Distribution of sympatric Microtus species at the trapping sites in Lithuania, 2018–2020. Due to small numbers of trapped M. agrestis and M. oeconomus, proportions of these species do not correspond to width of slices in the pie charts.
line 109 - Thus, at least four trapping sessions per site – there could be more than four? And does this mean than all plantations were pooled together?
Answer: we apologize mistake – there were exactly four sessions in the site, two habitats x two trapping sessions. We deleted “at least”.
Trappings were done separately in every site. Data were pooled just when analyzing habitats.
Comment: line 120 - Samples of hair for carbon and nitrogen stable isotope analysis were collected from 120 376 of the 509 trapped vole individuals – if hair was taken for analysis why such an invasive method (like snap traps and not live traps) was used? I think this should be somehow justified as in many countries this would be difficult to force.
Answer: from every individual, lungs, spleen, liver, kidneys, brain and muscle samples were taken for pathogen analysis; also, when dissecting, breeding data were recorded. All these issues are not related to the presented manuscript, but this perfectly justifies snap-trapping. And, please note, that in one habitat only few voles were trapped per year.
Also, orchard and plantation owners are trying to get rid from rodents by poisoning (using rodenticides), therefore our trapping did not pose bigger harm to the populations than that of poison.
Comment: line 127 – maybe it is worth saying how much hair was taken for analysis, this may be important for someone planning similar research but with live trapping
Answer: text added.
In fact, 3 mm wide (or even less) tuft should be enough, depending on the used elemental analyzer and an isotope ratio mass spectrometer.
Comment: line 179 - All of these co-trappings were in the apple orchards or control habitats – how about berries plantations? I guess with no clear description of the study sites and the whole study design it is difficult to understand results.
Answer: we added text, explaining co-trappings in the other habitats.
Lines 200-202 “In the plum orchards, and currant plantations only common voles were present, while in the raspberry plantations single individual of the root vole was trapped in addition to common voles (Table 2).”
Comment: line 181 - At the site level,.. – what do you understand by ‘the site level’?
Answer: we mean sympatric co-occurrence, that is, voles inhabit the same site, but not necessary the same habitat inhabited by all species. Text changed, now Lines 185 is
“Sympatric co-occurrence (the same site, but not necessary the same habitat) was more frequent… “
Comment: Table 2 – so all of those habitats were present at each trapping site? with the same number of traps for each habitats and the same trapping effort?
Answer: no, one agricultural habitat and one control habitat per site. Table 2 was supplemented with n, number of investigated habitats.
Number of traps was the same; two lines of 25 traps were set in small agricultural habitats, four lines – in large orchards or berry plantations, 1-2 lines – in meadows. However, this information is not related to the aim of this study, therefore not presented. Trapping details can be found in
Stirkė, V., Balčiauskas, L., & Balčiauskienė, L. (2021). Common Vole as a Focal Small Mammal Species in Orchards of the Northern Zone. Diversity, 13(3), 134.
Comment: Figure 2 – ok, here you refer to apple orchards, do this means that plantations were not treated together?; it should be said what were control habitats (e.g. in meadows put in brackets)
Answer: Figure 2 shows, how syntopic species of Microtus voles are distributed in the isotopic space (they are present in orchards and control meadows; we also merged these habitats together in Fig. 2a). In the Fig. 2 no data of the other habitats are included.
In the plantations and plum orchards only common vole was present (and 1 individual of the root vole) – position of these species is presented in the Fig. 4.
Reviewer 3 Report
Here the authors present a study comparing the isotopic/dietary niche of three Microtus species in Lithuania agricultural landscapes. The manuscript is interesting and presents relevant information on resource use and diet shifts by including stable isotope analysis. I found some inconsistencies in Methods and Results that would like the authors to respond to and made some suggestions that I hope will help the authors to improve the quality of the manuscript. Please see my comments below.
SIMPLE SUMMARY
It would be interesting to include the scientific name of the species analyzed to help readers that are not familiar with the species that occur in your study area.
L 12 – I suggest changing from ‘synonymous’ to ‘a proxy’ or other similar concepts since the isotopic niche does not represent or cover all dimensions of a dietary/trophic niche.
ABSTRACT
L 26 – Include the scientific name of the species.
INTRODUCTION
L 48-49 – Again, I think would be interesting for the readers to present the scientific and common name of the target species.
L 71 – What do you mean by ‘moderately antropogenic habitats’?
L 72 – Please explain which concept you adopted from Bolnick et al. 2010. It is not clear.
METHODS
Figure 1:
- I suggest placing a black contour around the study site where no vole species were captured to improve visibility, and also include it in the figure legend.
- The map is missing coordinates and scale, please, include them.
- If the proportions in the pie chart do not represent the exact proportion of individuals of each species, what do they represent?
- Complement the figure caption with the complete location of where your study was conducted.
L 109 – What do you mean by ‘in a concrete habitat’?
L 128-130 – Samples were cleaned before analysis? Aiming to remove residuals from the environment where the species were captured?
L 152-154 – I did not understand what you did here. You assessed the differences in the capture rate of each species among sites where two or three species were captured? Not clear. Why did you do this analysis? To determine which species was more abundant?
RESULTS
Overall – I suggest including the sample size and species scientific names in all figures, and also maintaining the same color pattern for the species among figures (e.g., colors are different in Figure 3).
L 173-183:
- This characterization of the captures must be included as an objective of the study.
- According to Figure 1, there are sites where all three species were captured, why it is not presented here?
Table 2:
- How different is the food resource availability at each habitat type? In Figure 2 you compare only apple orchards, meadows and irrespective of habitat type (grouping all results?). Did you discard samples from other habitat types? Or they are grouped? Not clear
- Could this difference in orchard and plantation types influence the values of the vole species, and as consequence altering the size and overlap of voles’ isotopic niches?
L 224-225 – It would be interesting presenting the overlap values among the isotopic niches.
Figure 3:
- I suggest using the same color pattern for all figures concerning species identification.
- It would be interesting to include the number of samples analyzed of each species at each panel
L 232-234 – These percentage values are very low, and apparently, different from what is presented in Figure 3.
L 238-240 – You should mention this in the Methods, or at least, in the first section of the results, linking it to the information in Table 2.
L 240-242 – I do not understand what do you mean by ‘cumulative influence of co-occurrence’.
L 242-246 – A mean difference of 0.65 ‰ (5.26 – 4.61) considering that δ15N values for common voles varied from 0.52 to 14.50; this is also true for the other two vole species that presented high variation in δ15N values.
Figure 4:
- There is only one individual of M. oeconomus in panel C?
- No individual of M. agrestis was analyzed in these habitats? It is mentioned in the figure legend, but I could not find it in the panels.
- My suggestion is to move this Figure to supplementary material.
DISCUSSION
- 289 – As in the ‘Simple summary’, I suggest changing from ‘synonymous’ to ‘a proxy’ or other similar concepts since the isotopic niche does not represent or cover all dimensions of a dietary/trophic niche.
L 301-303 – Results.
L 312 – Which species?
L 357 – Which authors?
Author Response
Rev#3 comments
Here the authors present a study comparing the isotopic/dietary niche of three Microtus species in Lithuania agricultural landscapes. The manuscript is interesting and presents relevant information on resource use and diet shifts by including stable isotope analysis. I found some inconsistencies in Methods and Results that would like the authors to respond to and made some suggestions that I hope will help the authors to improve the quality of the manuscript. Please see my comments below.
Answer: thank you very much for the positive evaluation of our manuscript. Answer to your comments are presented below.
SIMPLE SUMMARY
Comment: It would be interesting to include the scientific name of the species analyzed to help readers that are not familiar with the species that occur in your study area.
Answer: scientific names added as advised.
Comment: L 12 – I suggest changing from ‘synonymous’ to ‘a proxy’ or other similar concepts since the isotopic niche does not represent or cover all dimensions of a dietary/trophic niche.
Answer: text changed in Line 12 (proxy for …)
ABSTRACT
Comment: L 26 – Include the scientific name of the species.
Answer: scientific names added as advised.
INTRODUCTION
Comment: L 48-49 – Again, I think would be interesting for the readers to present the scientific and common name of the target species.
Answer: we do not refer to species in Lines 48–49. Therefore, scientific names added in the Lines 91–92, where we present hypotheses, and removed from Lines 103–104.
Comment: L 71 – What do you mean by ‘moderately antropogenic habitats’?
Answer: we had in mind, that agricultural habitats are just moderately anthropogenic, compared to territories of factories, mines, etc. We changed ‘moderately anthropogenic habitats’ to “agricultural habitats” to avoid misunderstandings
Comment: L 72 – Please explain which concept you adopted from Bolnick et al. 2010. It is not clear.
Answer: we added text to explain Bolnick opinion, now is “We therefore oriented to the evaluation of the trophic niche in agricultural habitats, considering also the opinion of Bolnick et al., [36] that ecological release from interspecific competition can lead to increases in niche width.”
METHODS
Figure 1:
Comment: I suggest placing a black contour around the study site where no vole species were captured to improve visibility, and also include it in the figure legend.
Comment: The map is missing coordinates and scale, please, include them.
Comment: If the proportions in the pie chart do not represent the exact proportion of individuals of each species, what do they represent?
Comment: Complement the figure caption with the complete location of where your study was conducted.
Answer: we accepted all four comments on the Figure 1. Study site with no Microtus voles was contoured black; we added scale bar and explanation on the disagreement of trapped numbers and slices in the pie charts for two species, which were trapped too scarcely to be shown according their share. These pie charts are intended to show presence of the species. We, however, do not see possibility to include coordinates of all sites into the map, as it will be much overcrowded. Therefore, coordinates of the country were added to the text. Changes are:
Figure 1. Distribution of sympatric Microtus species at the trapping sites in Lithuania, 2018–2020. Due to small numbers of trapped field and root voles, proportions of these species do not correspond to width of slices in the pie charts.
Voles were trapped at 18 study sites in Lithuania (northern Europe, 55°19’ N, 23°54’ E) in 2018–2020, covering agricultural habitats (orchards and berry plantations) and neighbouring meadows as control habitats.
Comment: L 109 – What do you mean by ‘in a concrete habitat’?
Answer: we mean “the same”, text changed
Comment: L 128-130 – Samples were cleaned before analysis? Aiming to remove residuals from the environment where the species were captured?
Answer: We wrote “The samples of the hair were not pre-treated, as earlier we ascertained that this procedure did not change the obtained results [55].” In the another study we compared stable isotope ratios using the same sample in two repeated analyses – first part of the sample was pre-treated (cleaned), second part was not cleaned. Results did not differ.
Therefore, we cleaned only samples dirty with soil or blood; if sample was very dirty, it was not analysed (was discarded). To explain, we added text at Line 133
“The samples of the hair were not pre-treated, as earlier we ascertained that this procedure did not change the obtained results [55]. Dirty (covered by soil or blood) samples were washed in deionized water and methanol, then dried. Very dirty samples were discarded.”
Comment: L 152-154 – I did not understand what you did here. You assessed the differences in the capture rate of each species among sites where two or three species were captured? Not clear. Why did you do this analysis? To determine which species was more abundant?
Answer: no no, this analysis is not related to the abundance of different vole species. We checked, how frequently two or three Microtus species were sympatric and/or syntopic. These results are presented in the Lines 178-188. As we use proportion (of the sites with three, two or single vole species), CI is the standard indicator.
RESULTS
Comment: I suggest including the sample size and species scientific names in all figures, and also maintaining the same color pattern for the species among figures (e.g., colors are different in Figure 3).
Answer: scientific names are already included.
We changed colors of Figures 1, 2 and 4 to match that of figure 3. We also add sample size to Figures 2, 4, 5 and 6.
L 173-183:
Comment: This characterization of the captures must be included as an objective of the study.
Answer: we changed aim as follows to incorporate form of co-occurrence (sympatric or syntopic).
“The aim of our study was to evaluate the width of the trophic niche of three sympatrically or syntopically co-occurring species of Microtus voles according to stable isotope (δ13C and δ15N) ratios.”
Comment: According to Figure 1, there are sites where all three species were captured, why it is not presented here?
Answer: in the revised version, text on sympatric co-occurrence is given in the Lines 204-207.
Sympatric co-occurrence (the same site, but not necessary the same habitat) was more frequent: at 22.2% (CI = 9.0–45.2%) of sites, common voles, field voles and root voles occurred in sympatry, and at 27.8% (12.5–50.9%) of sites two species co-occurred.
Table 2:
Comment: How different is the food resource availability at each habitat type? In Figure 2 you compare only apple orchards, meadows and irrespective of habitat type (grouping all results?). Did you discard samples from other habitat types? Or they are grouped? Not clear
Answer: unfortunately, analysis of the resources is just ongoing the phase of data collection, therefore cannot be presented in the manuscript. Preliminary, variation of resources is present even in the same habitats: for example, apple orchards are represented by young, old and medium age crops, intensity of agricultural activities are varying from low to high.
Therefore, to reflect this variation, we pooled data from all similar habitats (all apple orchards, all plum orchards, all currant plantations, all raspberry plantations, and all control meadows). Two of these habitats are presented in the Fig. 2, other ones, inhabited by common vole only – in the Fig. 4. So, we did not discard samples, just present results of analysis in different figures. This approach corresponds to our aim “evaluate the width of the trophic niche of three sympatrically or syntopically co-occurring species of Microtus voles according to stable isotope (δ13C and δ15N) ratios”
Comment: Could this difference in orchard and plantation types influence the values of the vole species, and as consequence altering the size and overlap of voles’ isotopic niches?
Answer: general answer is “yes”, of course isotopic space of voles depend on the resources, which may differ in the different habitats.
However, we present Figures 2–4, where we compare niches of different vole species within the same habitat groups, thus minimizing influence of the orchard and plantation types
Comment: L 224-225 – It would be interesting presenting the overlap values among the isotopic niches.
Answer: overlap percent was presented in the Lines 230-234; please have in mind, that overlap was calculated comparing overlap are with the sum of both core areas, therefore 4% or 7% or 9% are correct.
The overlap with the core dietary niche of the field vole in the control habitats (Fig. 3b) and with the root vole in the apple orchards (Fig. 3c) was less than 0.5%. The core dietary niches of the root and field voles widely overlapped in all cases: 4.3% irrespective of habitat (Fig. 3a), 7.2% in the control meadows (Fig. 3b) and 9.3% in the apple orchards (Fig. 3c).
Figure 3:
Comment: I suggest using the same color pattern for all figures concerning species identification.
Answer: color pattern in the Figures 1,2 and 4 was changed to match Figure 3.
Comment: It would be interesting to include the number of samples analyzed of each species at each panel
Answer: number of samples added to the caption
Comment: L 232-234 – These percentage values are very low, and apparently, different from what is presented in Figure 3.
Answer: overlap percent was calculated comparing overlap are with the sum of both core areas, therefore 4% or 7% or 9% are correct. Presentation in such a scale may yield visual impression, but hard data were used for calculations.
Comment: L 238-240 – You should mention this in the Methods, or at least, in the first section of the results, linking it to the information in Table 2.
Answer: comment accepted, information shown in Lines 201-203, with reference to the Table 2, as advised.
Comment: L 240-242 – I do not understand what do you mean by ‘cumulative influence of co-occurrence’.
Answer: apologies for mistype, word “cumulative” deleted.
Comment: L 242-246 – A mean difference of 0.65 ‰ (5.26 – 4.61) considering that δ15N values for common voles varied from 0.52 to 14.50; this is also true for the other two vole species that presented high variation in δ15N values.
Answer: ok, but this difference is 14.1% from the δ15N value in common voles without co-occurring other Microtus species, and is highly significant. We put 14.1% instead of the word “much”. Hope this is acceptable.
Figure 4:
Comment: There is only one individual of M. oeconomus in panel C?
Comment: No individual of M. agrestis was analyzed in these habitats? It is mentioned in the figure legend, but I could not find it in the panels.
Comment: My suggestion is to move this Figure to supplementary material.
Answer: yes, habitats shown in the Figure 4 are not characterized with presence of syntopic Microtus species. This is shown in the text, Lines266-268 “In the plum orchards (Fig. 4a) and currant plantation (Fig. 4b), we trapped only common voles, while in raspberry plantations (Fig. 4c) only a single root vole individual of was trapped along with a number of common voles.”
We changed figure legend.
Figure 4 is important, as it visualizes differences in isotopic space of M. arvalis, living in co-occurrence with the other species (Figure 2) and without co-occurrence (Figure 4). Placing both of these Figures in the main text, we seek to see these differences easier (note, that values of both axes differ). If Figure 4 would be in the Supplement, reader could not scroll the text to see them both.
DISCUSSION
Comment: L289 – As in the ‘Simple summary’, I suggest changing from ‘synonymous’ to ‘a proxy’ or other similar concepts since the isotopic niche does not represent or cover all dimensions of a dietary/trophic niche.
Answer: thank you, we used word “proxy”.
Comment: L 301-303 – Results.
Answer: we cannot present this chapter as Results – we are citing published paper, and using presented data as the only literature source to compare with results of investigation; therefore, it is Discussion.
Comment: L 312 – Which species?
Answer: field vole, mentioned in the previous sentence. We changed text to make this clear.
Comment: L 357 – Which authors?
Answer: J.M Bergeron and L. Joudoin; text changed as shown below.
J.M Bergeron and L. Joudoin pointed out, that diet changes under interspecific competition may be very important, as diet quality is related to survivorship and health status [77]. These authors wrote that herbivores are also limited by food sources and do have different food preferences.